# Toll-like Receptor Signaling Inhibitory Peptide Improves Inflammation in Animal Model and Human Systemic Lupus Erythematosus

**DOI:** 10.3390/ijms222312764

**Published:** 2021-11-25

**Authors:** Wook-Young Baek, Yang-Seon Choi, Sang-Won Lee, In-Ok Son, Ki-Woong Jeon, Sang-Dun Choi, Chang-Hee Suh

**Affiliations:** 1Department of Rheumatology, Ajou University School of Medicine, 164 Worldcup-ro, Suwon 16499, Korea; arikato83@naver.com (W.-Y.B.); znfdla@naver.com (S.-W.L.); 119iyla@naver.com (I.-O.S.); hortensia1225@gmail.com (K.-W.J.); 2Department of Molecular Science and Technology, Ajou University, 164 Worldcup-ro, Suwon 16499, Korea; chldidtjsdl@gmail.com (Y.-S.C.); sangdunchoi@ajou.ac.kr (S.-D.C.)

**Keywords:** lupus erythematosus, systemic, Toll-like receptors, mice, inbred MRL/*lpr*

## Abstract

Toll-like receptors (TLRs) play a major role in the innate immune system. Several studies have shown the regulatory effects of TLR-mediated pathways on immune and inflammatory diseases. Dysregulated functions of TLRs within the endosomal compartment, including TLR7/9 trafficking, may cause systemic lupus erythematosus (SLE). TLR signaling pathways are fine-tuned by Toll/interleukin-1 receptor (TIR) domain-containing adapters, leading to interferon (IFN)-α production. This study describes a TLR inhibitor peptide 1 (TIP1) that primarily suppresses the downstream signaling mediated by TIR domain-containing adapters in an animal model of lupus and patients with SLE. The expression of most downstream proteins of the TLR7/9/myeloid differentiation factor 88 (MyD88)/IFN regulatory factor 7 signaling was downregulated in major tissues such as the kidney, spleen, and lymph nodes of treated mice. Furthermore, the pathological analysis of the kidney tissue confirmed that TIP1 could improve inflammation in MRL/*lpr* mice. TIP1 treatment downregulated many downstream proteins associated with TLR signaling, such as MyD88, interleukin-1 receptor-associated kinase, tumor necrosis factor receptor-associated factor 6, and IFN-α, in the peripheral blood mononuclear cells of patients with SLE. In conclusion, our data suggest that TIP1 can serve as a potential candidate for the treatment of SLE.

## 1. Introduction

Systemic lupus erythematosus (SLE) is a chronic, multi-faceted autoimmune disease that usually manifests in various organs [1]. SLE-related immunomodulatory abnormalities induce overproduction of autoantibodies such as anti-nuclear antibodies (ANA) and anti-dsDNA antibodies, as well as immune complexes, and stimulate complement activation. The resulting tissue inflammation can mediate long-term damage [2,3]. While the exact cause of SLE is still unknown, genetic variation and environmental factors are thought be plausible factors [4,5]. Various genetic variations observed in patients with lupus, including Ets-1, have recently been identified, and immune profiling has been performed [6,7]. Thus, a combination of genetic defects and environmental factors such as hormones, ultraviolet light, drugs, and infections forces the immune system to fail to recognize the “self” and begin to attack cells and destroy various organs, leading to SLE [8].

Toll-like receptors (TLRs) are a family of pattern recognition receptors that play an important role in the initiation of innate immune responses by detecting potentially harmful pathogens [9]. TLR7 and TLR9 (also TLR3 and TLR8) can exclusively detect nucleic acid-type pathogens in endosomes [10] and are highly expressed in plasmacytoid dendritic cells (pDCs), which secrete high levels of interferon (IFN)-α and play pivotal roles in the innate and adaptive immune response switching [11]. Hence, the abnormal stimulation of TLR7 and TLR9 contributes to the pathology of autoimmune diseases such as SLE and rheumatoid arthritis (RA) [12]. TLR7 is known to play a pivotal role in autoantibody production in murine lupus and is localized within the intracellular endosome along with TLR3 and TLR9 [13]. The activation of TLR7 using a synthetic TLR7 ligand is known to stimulate the systemic production of IFN and aggravate lupus nephritis (LN) in MRL/MpJ-fas *lpr* (MRL/*lpr*, lupus-prone mice) [14]. TLR signaling pathway activation occurs in the cytoplasmic Toll/interleukin (IL)-1 receptor (TIR) domain, which binds to the TIR domain-containing adapter myeloid differentiation factor 88 (MyD88) [6]. MyD88 is a major TLR adapter [15] and dictates the response to TLR7 and TLR9 ligands [16]. Following TLR activation, a ‘Myddosome’ complex is formed between MyD88, IL-1 receptor-associated kinase 1 (IRAK1) and IRAK4, which initiates the signaling pathway, leading to the activation of transcription factors and the production of inflammatory cytokines [17]. IRAK family proteins then mediate MyD88-dependent TNF receptor-associated factor 6 (TRAF6) ubiquitination and downstream signaling in association with MyD88 and TRAF6. IRAK4 induces the phosphorylation of IRAK1, and phosphorylated IRAK1 dissociates from MyD88 and binds to TRAF6 [18]. IFN regulatory factor 7 (IRF7), activated by the ubiquitin ligase activity of TRAF6, translocates to the nucleus and mediates the transcription of type I IFN genes [19]. Thus, TLR7- and TLR9-mediated IFN-α induction is essential for the formation of complexes composed of MyD88, TRAF6, and IRF7 [20].

A recent study described a novel peptide, TLR inhibitory peptide 1 (TIP1), derived from the TIR domain-adapter protein that inhibited multiple TLR signaling pathways in murine and human cell lines [21]. The TIP1 peptide is a synthetic peptide (24 mer amino acids), which is a combination of cell penetrating peptide (Antennapedia cell penetrating peptide from *D. melanogaster*’s) and a mimetic sequence (from the TIR domain of human TIRAF). The direct interaction of TIP1 peptide to BB-loop in the TLR4 TIR domain was confirmed by a kinetic binding affinity test [21]. Treatment with TIP1 led to remarkable recovery in joint damage and inflammation markers in an animal model of RA [21]. Therefore, TIP1 may serve as a potential drug candidate for multiple TLR-mediated chronic inflammatory/autoimmune diseases.

In the present study, we hypothesized that TIP1 may ameliorate SLE symptoms by inhibiting TLR signaling complexes. We assessed the markers of SLE and TLR signaling factors after TIP1 treatment in MRL/*lpr* mice and peripheral blood mononuclear cells (PBMCs) of patients with SLE. We found that the levels of the markers of SLE were ameliorated following TIP1 treatment in MRL/*lpr* mice. Further, TLR signaling factors reduced after TIP1 treatment in both the major tissues of MRL/*lpr* mice and PBMCs of patients with SLE, suggestive of the TIP1-mediated alleviation of SLE.

## 2. Results

### 2.1. TIP1 Treatment Ameliorates Skin Damage in Lupus-Prone Mice (MRL/lpr)

The therapeutic efficacy of TIP1 was evaluated in MRL/*lpr* mouse, a well-known model of SLE characterized with high levels of circulating antibodies and inflammatory cytokines that develop an autoimmune disease similar to human SLE [22]. To evaluate its therapeutic potential, TIP1 was administered to B6J and MRL/*lpr* mice. As MRL/*lpr* mice spontaneously develop SLE over a few months, we started treating 14-week-old female MRL/*lpr* mice with an i.p. injection of TIP1 (10 nmol/g) five times per week for 4 weeks (Figure 1a). We captured dorsal and abdominal skin images to demonstrate any changes in the appearance of the whole mouse. TIP1 administration drastically reduced the skin damage in MRL/*lpr* mice, which showed healthier appearance than those treated with vehicle (Figure 1b).

### 2.2. TIP1 Treatment Reduces Nephromegaly, Splenomegaly, and Lymphadenopathy

We examined the size of the major organs from MRL/*lpr* mice treated with TIP1. The sizes of the kidney, spleen, and lymph node of MRL/*lpr* mice significantly increased. However, TIP1 administration greatly reduced the sizes of the kidney, spleen, and lymph node (Figure 1c–e). Thus, TIP1 appeared to be efficacious in reducing the progression of nephromegaly, splenomegaly, and lymphadenopathy in MRL/*lpr* mice.

### 2.3. TIP1 Treatment Decreases Anti-dsDNA Antibody and Urine Albumin Levels

We evaluated whether TIP1 can be useful for the treatment of SLE in a murine model by assessing its ability to reduce circulating levels of autoantibodies, a hallmark of SLE. We performed enzyme-linked immunosorbent assay (ELISA) for ANA and anti-dsDNA antibodies in the serum collected from the treated lupus-prone mice. The serum anti-dsDNA antibody titer was significantly lower in TIP1-treated mice than in vehicle-treated mice by 18 weeks of age (*p* < 0.005, Figure 2a). Although no significant difference was noted, ANA levels were also lower in TIP1-treated mice than in vehicle-treated mice (Figure 2b). However, no difference was observed in the serum levels of complement C3 (Figure 2c). TIP1-treated mice showed a significantly lower urine albumin level than vehicle-treated mice (Figure 2d). Together, these data show that TIP1 significantly reduced the levels of anti-dsDNA antibody and urine albumin after 4 weeks of daily dosing.

### 2.4. Therapeutic Effect of TIP1 in LN

To study whether TIP1 treatment ameliorates LN, we performed histological analysis of the kidney tissue. Renal histopathologic features were determined by hematoxylin and eosin (H&E) and periodic acid–Schiff (PAS) staining in each group of mice, and the severity of LN was assessed in a blinded manner. Mesangial cell population, endothelial cell proliferation, and inflammatory cell infiltration around the glomeruli increased in the mice from the vehicle-treated group, but significantly decreased in TIP1-treated mice. The size of the glomeruli significantly increased in control mice, but decreased by approximately 30% following TIP1 treatment (Figure 3a).

### 2.5. TIP1 Inhibits the TLR7/9 Signaling Pathway

TLR7/9 binds to the adapter protein MyD88, which induces IRAK1 phosphorylation via IRAK4. IRAK1 interacts with TRAF3/6 and activates the IRF7 signaling pathway [23]. To investigate whether TIP1 treatment inhibits the TLR7/9/MyD88/IRF7 signaling pathway, we analyzed protein expression in major tissues and found that the expression of MyD88 and IRAK4 was significantly lower in the kidney tissues of TIP1-treated mice than in those of vehicle-treated mice (Figure 3b,c). In the spleen, the levels of TLR7, MyD88, IRAK4, and TRAF6, but not TRAF3, significantly decreased after TIP1 treatment (Figure 4a,b). The decrease in protein expression, including TRAF3 expression, in the spleen following TIP1 treatment was also observed by immunohistochemistry (Figure 4c). Next, we examined changes in TLR7/9/MyD88/IRF7 signaling pathway-related proteins in the lymph node and found that the levels of TLR9, MyD88, and TRAF6 significantly decreased in the lymph node of TIP1-treated mice (Figure 5a,b).

### 2.6. Inhibitory Effects of TIP1 on TLR Signaling Pathways Ex Vivo

Considering the decrease in the expression of the proteins related to the TLR signaling pathway following TIP1 treatment in MRL/*lpr* mice (Figure 3, Figure 4 and Figure 5), we investigated any alterations in their levels in the PBMCs from patients with SLE. We assessed MyD88/IRAKs/TRAFs/IRF7/IFN-α protein expression levels in TIP1-treated cultured PBMCs from patients with SLE and age- and sex-matched healthy subjects (HSs). We found that the expression of TLR signaling proteins decreased after TIP1 treatment in PBMCs from patients with SLE (Figure 6a,b).

We noted improvement in clinical manifestation and autoantibody production in MRL/*lpr* mice treated with TIP1. TIP1 peptide reduced the expression of the proteins related to TLR signaling pathway, not only in MRL/*lpr* mice but also in the cultured PBMCs of patients with SLE.

## 3. Discussion

SLE is a heterogeneous autoimmune disease involving multiple organ systems [24]. The management and treatment of SLE is complex and performed in different ways. To date, antimalarials, corticosteroids, immunosuppressants, and biologics have been tested to treat SLE, and nonsteroidal anti-inflammatory drugs have been used for the treatment of inflammation and pain [25]. While SLE treatment has evolved over time, efforts are underway to develop new disease-specific drugs. It is important to develop treatment regimens for lupus with no or minimum side-effects, and studies have been directed to find new targets. In this direction, recent studies have focused on targeting TLRs and modulating TLR signals. TIP1 has been recently developed by Choi’s group, and its efficacy has been demonstrated both in vitro and in animal models of RA [21].

In MRL/*lpr* mice, apoptosis is suppressed owing to the absence of Fas signaling, resulting in lupus-like symptoms [26]. MRL/*lpr* mouse is a well-known lupus model that shows high levels of autoantibodies in the serum at 6 weeks of age. MRL/*lpr* mouse exhibits accelerated lymphadenopathy from 12 weeks of age and LN at 12–16 weeks of age. Approximately 50% of lethality is attributed to uremia observed until 20–22 weeks of age. In general, all animals die by 28 weeks [27,28]. To determine whether TIP1 peptide could improve SLE condition, we investigated the efficacy of a 4-week TIP1 treatment regimen in 14-week-old MRL/*lpr* mice. We compared the overall phenotype between vehicle-treated and TIP1-treated mice and found severe skin diseases and enlarged major tissues such as the spleen, lymph node, and kidney in control mice. However, skin diseases were rarely observed and the sizes of the major tissues reduced in TIP1-treated mice.

Autoantibodies such as ANAs and anti-dsDNA antibodies serve as the hallmark of SLE and are one of the main diagnostic criteria [29]. More than 95% of patients with SLE exhibit ANAs, and anti-dsDNA antibodies have been associated with the disease activity of SLE and LN [30]. In addition, urinary albumin concentration increases with the elevation in the disease activity in the kidney. Thus, urinary albumin can serve as a biomarker for LN [31,32]. Complement is associated with the pathogenesis of SLE, and in particular, tissue damage in SLE occurs following the activation of complement and production of immune complexes [33,34]. Therefore, we verified whether TIP1 treatment can improve lupus symptoms by monitoring various indicators of SLE. We measured autoantibodies such as anti-dsDNA in the serum and analyzed albumin levels in the urine, well-known indicators of SLE disease activity. We found that serum autoantibodies significantly increased in vehicle-treated mice but reduced following TIP1 treatment. Albumin concentration also significantly decreased in the urine of TIP1-treated mice. The levels of complement C3 were slightly higher in the mice treated with TIP1 than in the control mice; however, this difference was not significant. These results demonstrate that TIP1 ameliorates some SLE symptoms and key indicators of disease activity in the serum and urine.

A serious complication of SLE is LN; more than 50–60% patients with SLE are known to have kidney involvement within 10 years of diagnosis [35]. LN is a severe complication characterized with the production of anti-dsDNA antibodies owing to the deposition of immune complexes in the kidney, leading to glomerular damage [36]. TLR3, TLR7, and TLR9 were found to be overexpressed in the kidney of patients with LN and their levels correlated with clinical and histologic indications [37]. In LN, high titers of anti-dsDNA antibodies and decreased complement levels were associated with inflammation progression and kidney damage [38,39]. In this study, histopathological analysis confirmed recovery after TIP1 treatment compared to the condition after vehicle treatment in mice with advanced kidney tissue damage. Damaged kidney tissues in control mice were associated with increased anti-dsDNA antibody levels in the serum and elevated albumin concentrations in the urine. The amelioration of LN in MRL/*lpr* mice following TIP1 treatment was consistent with the decrease in anti-dsDNA antibody and urinary albumin concentrations. Thus, TIP1 treatment alleviated major SLE symptoms and the pathology of LN.

More than 40% patients with SLE show defects in the clearance of apoptotic cells, resulting in poor elimination of dead cells by phagocytes. The incomplete clearance of these cellular debris increases the release of host RNA and DNA, which are detected by TLRs to induce the production of autoantibodies [40]. Patients with SLE have increased blood levels of IFN-α, which are associated with disease activity [41]. In addition, immune complexes containing RNA isolated from the serum of patients with SLE stimulate pDCs to produce IFN-α. TLR7 is involved in the development of SLE upon activation by RNA-containing immune complexes and is important for SLE pathogenesis through participation in inflammatory signal transduction [40]. Recent studies have shown increased signaling through TLR7 in SLE and the elevated mRNA expression of TLR7 and TLR9 in PBMCs of patients with SLE [42]. To date, genome-wide association studies, in vitro studies, experimental mouse models, and clinical sample analysis have provided relevant evidence on the involvement of TLRs, including TLR7/9, in SLE onset. Therefore, targeting TLRs and regulating TLR signaling have emerged as important strategies for the treatment of SLE [32,43,44]. The innate immune system recognizes damage-associated molecular patterns (DAMPs), which are nucleic acid-containing debris released from dead cells. TLRs interact with MyD88- and TIR domain-containing adaptor protein-inducing IFN-β-mediated pathway molecules while detecting DAMP [9,21,45,46]. The induction of type 1 IFN expression by TLR7 and TLR9 relies entirely on MyD88 in pDCs. Activation of TLR7 or TLR9 in pDC recruits MyD88 after IRAK4 recruitment, and the MyD88 complex contains TRAF3, TRAF6, IRAK4, IRAK1, etc. [47,48]. IRAK1 exhibits a TRAF6-binding site and may contribute to TRAF6 activation [49]. As IRAK1 can directly bind and phosphorylate IRF7, it acts as an IRF7 kinase and mediates IFN-α induction downstream of MyD88 and IRAK4 [23,24]. We evaluated the expression of the proteins involved in TLR7/9 signaling in major tissues such as the kidney, spleen, and lymph node and found them to be downregulated following TIP1 treatment. Similar results were observed in the PBMCs of patients with SLE. Thus, many downstream proteins of the TLR signal transduction cascade (such as MyD88, IRAK, TRAF, and IFN-α) may be considered as potential therapeutic targets for SLE treatment.

In summary, TIP1 ameliorates autoantibody production and disease activity in a murine lupus model. The pathological analysis of the renal tissue of MRL/*lpr* mice reveals the TIP1-mediated improvement in the immunopathology of LN. The endosomal TLR7/9 signaling pathway is controlled by TIP1, which suppresses the MyD88 receptor and downstream molecules. Our clinical data show that PBMCs from patients with SLE exhibit reduced endosomal TLR7/9/MyD88/IRF7 signaling following TIP1 treatment. In conclusion, the TIP1 can be a plausible candidate for the treatment of SLE and LN.

## 4. Materials and Methods

### 4.1. Animals

All animal experiments were compliant with the ARRIVE guidelines and were carried out in accordance with the U.K. Animals (Scientific Procedures) Act, 1986 and associated guidelines, EU Directive 2010/63/EU for animal experiments, or the National Institutes of Health guide for the care and use of Laboratory animals (NIH Publications No. 8023, revised 1978). All animal procedures were reviewed and approved by the animal ethics committee of Ajou University Medical Center (Approval No. 2017-0022). Female C57BL/6J (B6J, wild-type, 19–20 g) and MRL/*lpr* (36–40 g) mice were purchased from the Jackson Laboratory (Bar Harbor, ME, USA). Mice were allowed to acclimatize for 1 week before experimentation and were maintained under pathogen-free conditions according to the guidelines of the animal facility issued by Ajou University School of Medicine. For experiments, mice from the control group received an intraperitoneal (i.p.) injection of vehicle (phosphate-buffered saline (Thermo Fisher Scientific, Waltham, MA, USA), 137 mM NaCl, 2.7 mM KCl, 8.1 mM NaH2PO4, 1.47 mM KH2PO4), while those from the TIP1 treatment group received a single daily i.p. injection of 10 nmol/g TIP1 for 4 weeks from 14 to 18 weeks of age. Each group had five animals. During the treatment period, mice were weighed once a week. At the end of the experimental period, mice were sacrificed, and their blood, urine, and organs were collected. Blood samples were left undisturbed for 1 h, and then collected into serum separation tubes and centrifuged at 1500× *g* for 10 min at 4 °C. Serum samples obtained were stored at −80 °C. Collected urine samples were immediately stored at −80 °C. SLE-targeted organs (the kidney, spleen, and lymph node) were rapidly excised, and tissue of average size of kidney, spleen, and lymph nodes from each group of mice was photographed as representative, thoroughly washed with PBS, and immersed in an RNA stabilization solution (QIAGEN Sciences, Maryland, MD, USA).

### 4.2. Measurement of Disease Markers of SLE

The concentration of autoantibodies and activity markers in the mouse serum was analyzed by ELISAs specific for mouse anti-dsDNA antibody (Mybiosource, San Diego, CA, USA), mouse ANA (Mybiosource, San Diego, CA, USA), and mouse C3 complement (Mybiosource, San Diego, CA, USA). To examine kidney functions, urine samples were collected before mice were sacrificed. Urinary levels of albumin were analyzed using a mouse albumin ELISA kit (Alpco Diagnostics, Salem, NH, USA) according to the manufacturer’s protocols. Three independent measurements per sample were recorded.

### 4.3. Histology and Immunohistochemistry of the Kidney Tissue

The kidney, spleen, and lymph node tissues were fixed in 4% paraformaldehyde at 4 °C for overnight. The fixed tissues were paraffin-embedded, and 2–4 μm kidney sections were stained with H&E or PAS reagents. Immunodetection was performed using Rabbit polyclonal antibody (pAb) to TLR7 (Abcam, Cambridge, MA, USA), Rabbit pAb to TLR9 (Abcam, Cambridge, MA, USA), Rabbit pAb to TRAF3 (LSBio, Seattle, WA, USA), Rabbit pAb to TRAF6 (LSBio, Seattle, WA, USA), Rabbit pAb to IRAK4 (LSBio, Seattle, WA, USA), Rabbit pAb to MyD88 (LSBio, Seattle, WA, USA), and Rabbit pAb to IL-17 (Abcam, Cambridge, MA, USA) antibodies as per the standard protocol. Briefly, 4 μm sections of 4% paraformaldehyde (PFA)-fixed paraffin-embedded blocks were rehydrated and blocked with 3% bovine serum albumin at 20–25 °C for 1 h. The sections were incubated with each primary antibody for 30 min at 20–25 °C. After incubation, the slides were washed and probed with a secondary anti-rabbit IgG antibody (Polyview Plus HRP-DAB Kit, Enzo Lifesciences., Farmingdale, NY, USA) at 20–25 °C for 1 h and treated with the 3,3′-diaminobenzidine substrate. Counterstaining was performed using methyl green.

### 4.4. Western Blot Analysis

Total protein extraction was performed using radioimmunoprecipitation assay buffer (iNtRON Biotechnology, Seongnam, Gyeonggi-do, Korea). Equal amounts of proteins were resolved by sodium dodecyl sulfate–polyacrylamide gel electrophoresis and analyzed using Rabbit monoclonal antibody to TLR7 (Cell Signaling Technology Inc., Danvers, MA, USA), Rabbit pAb to TLR9 (Abcam, Cambridge, MA, USA), Rabbit pAb to MyD88 (LSBio, Seattle, WA, USA), Rabbit pAb to IRAK4 (LSBio, Seattle, WA, USA), Rabbit pAb to TRAF3 (LSBio, Seattle, WA, USA), Rabbit pAb to TRAF6 (LSBio, Seattle, WA, USA), Rabbit pAb to IRF7 (LSBio, Seattle, WA, USA), Rabbit pAb to IFN-α (Thermo Fisher Scientific, Waltham, MA, USA), and polyclonal goat anti-cytoskeletal actin (Bethyl, Montgomery, TX, USA). Density analysis was performed using ImageJ analysis software. (Version 1.52a, National Institutes of Health, Bethesda, MD, USA).

### 4.5. Ex Vivo Cell Culture

All patients with SLE satisfied at least four of the revised American College of Rheumatology classification criteria [50]. Age- and sex-matched HSs with no history of autoimmune disorders were also enrolled. The study was approved by the Ajou Institutional Review Board (Approval No. AJIRB-BMR-OBS-15-340), and informed consent was provided by all subjects. The study procedures were carried out according to the Declaration of Helsinki and Good Clinical Practice guidelines.

The blood samples from patients with SLE were collected in BD Vacutainer CPT Mononuclear Cell Preparation Tubes (BD Biosciences, San Jose, CA, USA) supplemented with sodium citrate anticoagulant and immediately centrifuged. The uppermost plasma layer was discarded, while PBMCs were separated from the buffy coat. PBMCs were washed twice with PBS by centrifugation and cultured for 1 day at 1.5 × 10^6^ cells/mL in six-well plates containing Roswell Park Memorial Institute-1640 medium (Gibco by Life Technologies, Grand Island, NY, USA) with 10% Fetal Bovine Serum (Gibco by Life Technologies, Grand Island, NY, USA). On day 2, cells were incubated with TIP1 peptide (50 μM) for 24 h. Cells were then harvested by centrifugation, and the pellets were resuspended in PBS and subjected to protein extraction for Western blot analysis.

### 4.6. Data Analysis

Statistical analyses were performed using SPSS statistical software (Version 18, SPSS Inc., Chicago, IL, USA). The data are shown as means ± standard deviation or median and interquartile range, as appropriate. Differences in cytokine levels were determined using the Mann–Whitney *U* test. A value of *p* < 0.05 was considered statistically significant.

## Figures and Tables

**Figure 1 ijms-22-12764-f001:**
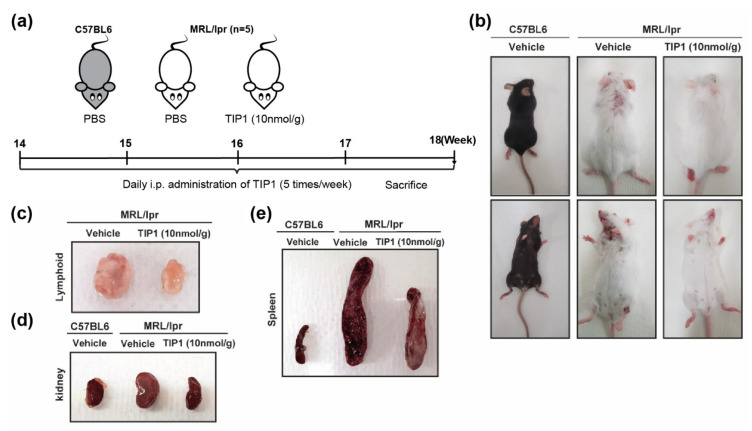
The dramatic inhibitory effect of Toll-like receptor inhibitor peptide 1 (TIP1) on systemic lupus erythematosus (SLE) in a mouse model. (**a**) Summary of the experimental validation of the inhibitory effect of TIP1 on SLE in a lupus-prone mouse. (**b**) Images of the whole body of female lupus-prone mice (MRL/*lpr*) and age- and sex-matched controls (C57BL/6J). Ameliorative effects of TIP1 on inguinal lymphoproliferation (**c**), nephromegaly (**d**), and splenomegaly (**e**). PBS, phosphate-buffered saline. n = 5 mice/group.

**Figure 2 ijms-22-12764-f002:**
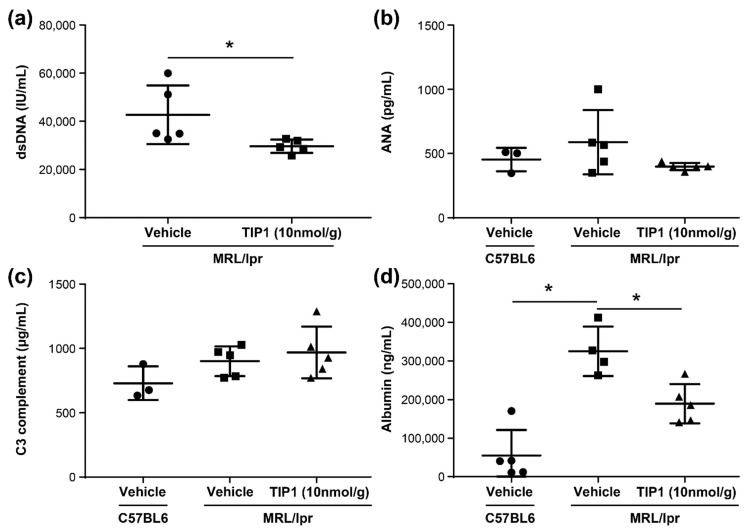
(**a**) Anti-dsDNA antibodies, (**b**) anti-nuclear antibodies (ANA), (**c**) C3 complement levels in the serum, and (**d**) albumin content in the urine as determined by enzyme-linked immunosorbent assays (ELISAs). All experiments were performed in duplicate wells (n = 4–5 mice/group). The exact Mann–Whitney U test was performed to compare the mean values between groups. * *p* < 0.05. TIP1, Toll-like receptor inhibitor peptide 1.

**Figure 3 ijms-22-12764-f003:**
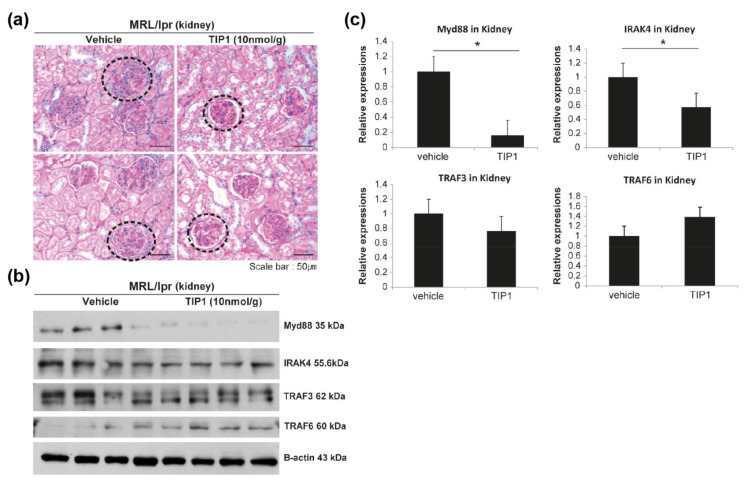
Toll-like receptor inhibitor peptide 1 (TIP1) improved nephritis in lupus-prone mice. (**a**) Periodic acid–Schiff (PAS) staining in the kidneys treated with TIP1. Representative microphotographs from PAS staining of the kidney tissues showing mesangial cell population, endothelial cell proliferation, and inflammatory cell infiltration around the glomerulus of lupus-prone mice (MRL/*lpr*; left panel, vehicle group; right panel, TIP1-treated group). n = 5 mice/group. (**b**) The total expression levels and (**c**) quantitative densitometry of endosomal Toll-like receptor signaling proteins in kidney tissues (n = 4 mice/group). * *p* < 0.05 by Mann–Whitney *U* test. The experiments were repeated at least twice. MyD88, myeloid differentiation factor 88; IRAK4, interleukin-1 receptor-associated kinase 4; TRAF3, tumor necrosis factor receptor-associated factor 3; TRAF6, tumor necrosis factor receptor-associated factor 6.

**Figure 4 ijms-22-12764-f004:**
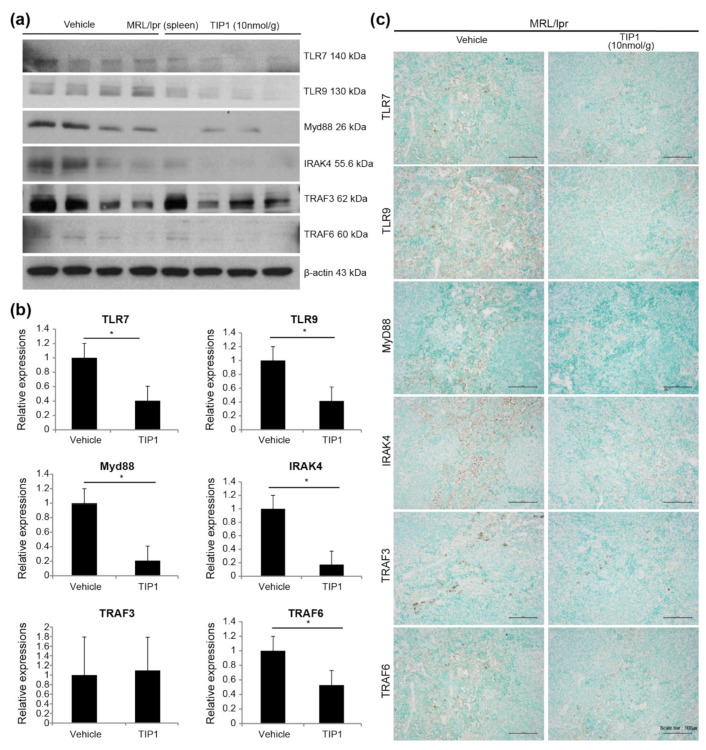
Toll-like receptor inhibitor peptide 1 (TIP1) blocks the endosomal Toll-like receptor 7 (TLR7)/TLR9 signaling pathway in the spleen. (**a**) The expression of the proteins of the endosomal TLR signaling was analyzed by Western blotting using spleen tissues from female lupus-prone mice (MRL/*lpr*). (**b**) The summarized bar graph shows the band intensity presented as the ratio of the target protein to actin (n = 4 mice/group). Values are means ± standard deviation (SD). * *p* < 0.05 by Mann–Whitney *U* test. (**c**) Immunohistochemistry was performed to detect endosomal TLR pathway proteins (brown) in the spleen (magnification, ×20). The experiments were repeated at least twice. MyD88, myeloid differentiation factor 88; IRAK4, interleukin-1 receptor-associated kinase 4; TRAF3, tumor necrosis factor receptor-associated factor 3; TRAF6, tumor necrosis factor receptor-associated factor 6.

**Figure 5 ijms-22-12764-f005:**
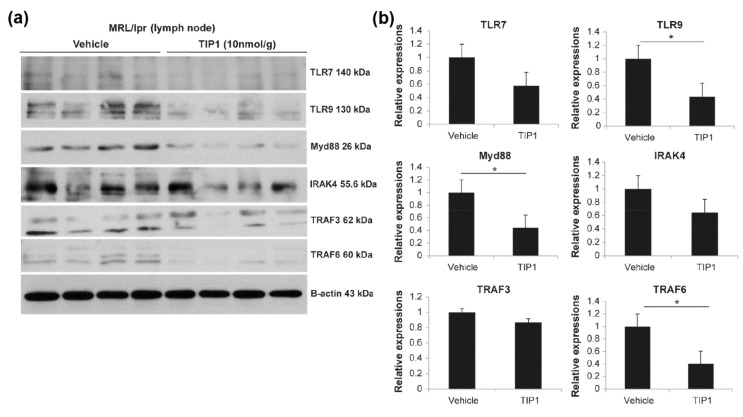
Inhibition of Toll-like receptor 7(TLR7)/TLR9 results in the blockade of the endosomal TLR signaling pathway in the lymph nodes. (**a**) The expression of total proteins related to the endosomal TLR signaling was analyzed by Western blotting using lymph nodes from female lupus-prone mice (MRL/*lpr*). (**b**) Western blot data were quantified using a densitometer (**P* < 0.05, Mann–Whitney *U* test). n = 4 mice/group. The experiments were repeated at least twice. TIP1, Toll-like receptor inhibitor peptide 1; MyD88, myeloid differentiation factor 88; IRAK4, interleukin-1 receptor-associated kinase 4; TRAF3, tumor necrosis factor receptor-associated factor 3; TRAF6, tumor necrosis factor receptor-associated factor 6.

**Figure 6 ijms-22-12764-f006:**
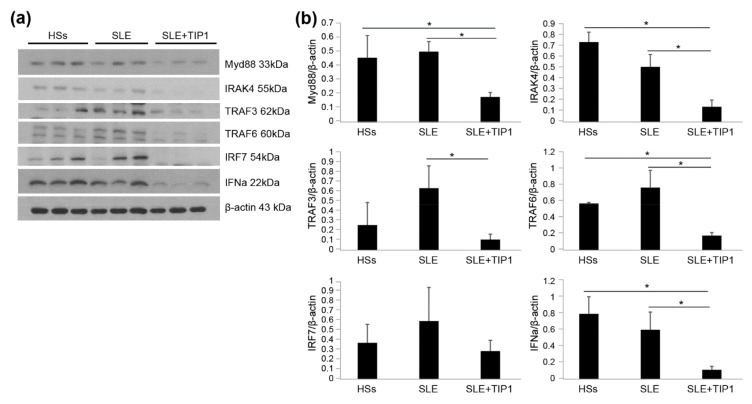
Inhibition of the Toll-like receptor (TLR) signaling pathway in peripheral blood mononuclear cells (PBMCs) of patients with systemic lupus erythematosus (SLE). (**a**) The expression of the proteins related to the TLR signaling was analyzed by Western blotting using PBMCs from patients with SLE (n = 3) and healthy subjects (HSs; n = 3). All 3 SLE patients were taking hydroxychloroquine 200 mg/d or 300 mg/d. Two SLE patients were not taking corticosteroids except one SLE patient taking prednisolone 2.5 mg/d. (**b**) Western blot data were quantified using a densitometer (* *p* < 0.05, Mann–Whitney *U* test). The experiments were repeated at least twice. TIP1, Toll-like receptor inhibitor peptide 1; MyD88, myeloid differentiation factor 88; IRAK4, interleukin-1 receptor-associated kinase 4; TRAF3, tumor necrosis factor receptor-associated factor 3; TRAF6, tumor necrosis factor receptor-associated factor 6; IRF7, interferon regulatory factor 7; IFN-α, interferon-α.

## Data Availability

The data presented in this study are available on request from the corresponding author.

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
