# Peer review of "Toll-like Receptor Signaling Inhibitory Peptide Improves Inflammation in Animal Model and Human Systemic Lupus Erythematosus"

_ijms, 2021, doi:10.3390/ijms222312764_

Round 1
Reviewer 1 Report
The article is an original and interesting research. I suggest to modify the presentation in the sections: introduction, materials and methods, discussions and conclusions and statements by the authors. The conclusions need to be presented more
Author Response
Thank you for your positive comment. We followed the format of the article. We changed the conclusion as below and and marked the changes by red-coloured text.
In conclusion, the TIP1 can be a plausible candidate for the treatment of SLE and LN.
Reviewer 2 Report
In this manuscript Authors continue to investigate a specific decoy peptide of toll-like receptors (TLR) named TIP1. Benefits of Myd88 cascade inhibition by TIP1 were described previously by the same Korean group in experimental LPS induced sepsis, collagen or carrageenan induced arthritis or ischemia-reperfusion injury of the kidney. In the current study a spontaneous mice strain (MRL/lpr) was used prone to lupus-like autoimmunity due to the mutation of Fas receptor. Similarly to the previously reported results, administration of TIP1 was protective against development of symptoms of lupus-like autoimmune pathology. Authors also presented inhibition of Myd88 pathway in human polymorphonuclear leucocytes isolated from patients with systemic lupus erythematodes. The experiments are clearly presented and well documented by graphical data presentations. Thus, I have only minor comments to this highly relevant manuscript:
- Results presented in figures 2 and 3 have a comment in their legend that all experiments were repeated at least thrice. Does it mean that the material was collected from 3 or more animals?
- In TIP1 treated animals Myd88 apparently disappears from the spleen or kidney. How this could be explained by Authors experience with the peptide. Is that regulated by proteasome tagging?
- What is the proof for endosomal localisation of TLR7/9 or TRAFs in the spleen sections (Fig. 4c). Under the low magnification it cannot be even attributed to any specific spleen cells.
- How Authors would explain a discrepancy between lpr mouse and human SLE leucocytes in relative changes of Myd88 vs. its downstream effectors. In human cells, there is only a moderate decrease of Myd88 and a profound of IRAKs or TRAFs, which is on the contrary to the animal model
- Sentence on the page 9 lines 252-253 should be corrected: immune-complexes simulate DCs to produce IFNa, while those containing RNA stimulate to produce IFNa?
- On the page 10 a recipe for PBS (lines 297-298) is wrong, this is 10 times PBS stock solution.
Author Response
We attached our response.

Reviewer 3 Report
In this study, the therapeutic effect of an inhibitor of TLR signaling, namely TLR inhibitor peptide 1 (TIP1), is examined in the MRL/lpr mouse model of systemic lupus erythematosus (SLE). TIP1 treatment had promising results in reducing skin lesions, lymphoid, spleen and kidney size and inflammation in MRL/lpr mice. These findings are really interesting and with high therapeutic value; however, the figures provided do not fully support the authors’ claims and conclusions. Specifically, the following minor and major points need to be clarified:
- In general, a clearer and more detailed description of the findings illustrated in each picture is needed.
- Figure 1A: the fact that the administration of TIP1 starts in the 14th week of mice is not shown.
- Figure 2: The levels of anti-dsDNA antibodies in C57BL6 mice are not shown. Are they negative? In this context, 400-500 pg/ml of ANA are detected in C57BL6 mice (Figure 2B). How the authors explain the presence of ANA in control mice? If this represents the cut-off of the detection ELISA and its negative, it should be clearly stated in figure legend and materials and methods section. Since the distribution of levels seem to be highly heterogeneous in some cases, data should be presented in dot blots to enable readily evaluation of sample distribution.
- Figure 3: A double band is detected in TRAF3 protein (Figure 2B, third panel). Which one corresponds to the 62kDa TRAF3 protein and how the authors ensured the specificity of staining (positive and negative controls are missing)? Although the protein estimation based on immunoblotting gives some information about the differential expression of the TLR signaling related molecules in kidney tissues, immunohistochemical detection of these molecules is needed for the identification of the type and the number of expressing cells.
- Similarly to TRAF3 in Figure 3, double bands are shown in TLR7, TLR9 and TRAF6 (but not TRAF3) panels of Figure 4A. Better, representative immunoblotting figures should be used with positive and negative controls. Furthermore, the specificity of the immunohistochemical stainings presented in Figure 4C is under question, since staining is not confined into cells; it seems like a diffuse non-specific uptake on the extracellular matrix. The presentation of panels inside each photograph that display higher magnification of the cellular stainings would be helpful. Again negative controls are missing.
- Figure 5: again multiple bands are shown in the immunoblotting detection of TLR7, TLR9, IRAK4, TRAF3 and TRAF6 questioning the specificity of detection. Immunohistochemical detection of TLRs and TLR signaling related molecules is also needed.
- Figure 6: the multiple bands are shown in the immunoblotting detection of TRAF3 and TRAF6 in this figure. Expression of TLR7 and TLR9 should also be presented.
Author Response
We attached our response.

Round 2
Reviewer 3 Report
The manuscript in its present form is appropriate for publication.